

# Phase transitions in quantum dot-Majorana zero mode coupling systems

Yue Mao[1] and Qing-Feng Sun[1,2]⋆

**1** International Center for Quantum Materials, School of Physics,
Peking University, 100871 Beijing, China
**2** Hefei National Laboratory, Hefei, 230088 Anhui, China

⋆ sunqf@pku.edu.cn

## Abstract

The magnetic doublet ground state (GS) of a quantum dot (QD) could be changed to a spin-singlet GS by coupling to a superconductor. In analogy, here we study the GS phase transitions in QD-Majorana zero mode (MZM) coupling systems: GS behaves phase transition versus intra-dot energy level and QD-MZM coupling strength. The phase diagrams of GS are obtained, for cases with and without Zeeman term. Along with the phase transition, we also study the change of spin feature and density of states. The properties of the phase transition are understood via a mean-field picture. Our study not only serves as an analogue to QD-superconductor phase transitions, but also gives alternative explanations on MZM-relevant experiments.

| | |
|---|---|
| Received | 2024-06-12 |
| Accepted | 2025-02-26 |
| Published | 2025-03-11 |

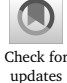

# 1   Introduction

When a quantum dot (QD) couples to a BCS-type superconductor, rich physical contents emerge in the quantum phase transition of the QD [1–7]. By controlling the intra-dot energy level, the QD itself could exhibit two kinds of ground states (GSs): a magnetic doublet state and a spin singlet state. The doublet state represents two degenerate spin-$\hbar/2$ states, a spin-up state $|\uparrow\rangle$ and a spin-down state $|\downarrow\rangle$. The QD is occupied by one electron, while the level with opposite spin is repulsed above Fermi surface by the Coulomb interaction and is empty. The singlet state originates from spinless states $|0\rangle$ and $\frac{1}{\sqrt{2}}(|\uparrow\downarrow\rangle-|\downarrow\uparrow\rangle)$, with zero and two electrons occupied, respectively. When coupled to a superconductor, the doublet state of the QD could be changed to a singlet state either by the proximity effect of spin-singlet Cooper pairs or by coupling to the quasiparticles outside the gap [6,8–10]. Whether the GS is doublet or singlet is mostly determined by the charging energy, the intra-dot energy level, and the coupling strength [2–7, 11, 12]. This doublet-singlet phase transition plays an important role in properties of the QD-superconductor hybrid devices, such as $0-\pi$ transition of Josephson junctions [1, 13, 14] and level crossing of Andreev bound states [2–6].

In certain superconducting systems, there could exist a special Andreev bound state called Majorana zero mode (MZM), which is its own antiparticle [8, 9, 15–37]. MZM is a hotspot in condensed matter physics because of its non-Abelian statistics, which can be managed to achieve fault-tolerated topological quantum computation [38–41]. Like a superconductor, the MZM also couples to electron and hole simultaneously [42]. Especially, because of its self-Hermitian property, the half fermionic MZM couples to a certain spin channel, leading to the resonant equal-spin Andreev reflection [26, 27, 29, 43, 44]. The MZM thus behaves strong spin-triplet pairing correlations [44,45], and induces a zero bias peak spectrum in both charge transport and spin-dependent transport [42, 43, 46].

In platforms for generating MZMs, Coulomb interaction could play an important role by influencing the Andreev bound states [4, 6, 9, 21, 24–27]. In particular, a QD region can be formed nearby the MZM, e.g. by an adatom deposited on the iron-based superconductor [9,47] or by a section of the Majorana nanowire [4–6, 21]. The QD-MZM coupling system can be regarded as a counterpart to the QD-superconductor hybrid structure, because the MZM is an Andreev bound state generated by the superconductor. But differently, the coupling term between the QD and the MZM involves only one spin channel, destroying the spin rotation symmetry. Compared to coupling with a conventional superconductor, does phase transition also happen in QD-MZM coupling systems? Will the peculiar features of the MZM lead to novel transition characteristics?

In this paper, we study the QD-MZM coupling system and find the corresponding phase transitions. Because spin rotation symmetry is broken, the degeneracy of the magnetic doublet state is destroyed, with GS becoming a spin-polarized state. By changing the intra-dot energy level and coupling strength, phase transition of GS happens with spin reversed. We study two cases without and with Zeeman term (which should be included considering experimental conditions), and give global phase diagrams showing the phase transition lines. These phase transitions influence occupation numbers, spin polarization, density of states (DOS), and the weight of zero energy state. These features are explained by a mean-field picture. Our theoretical results are also discussed by comparing with experimental observations. These phase transitions can provide an insight on MZM-related transport experiments.

The rest of this paper is as follows: In Sec. 2, the model and formula of the system are given. In Sec. 3, we study the phase transitions without Zeeman term. In Sec. 4, we consider the Zeeman term and study the corresponding phase transitions. At last, a brief conclusion is given in Sec. 5. In Appendix A, we explain the role of normal lead in detail.

## 2 Model and formula

As shown in Fig. 1, the system we study consists of a QD coupled to a MZM and a normal lead. The total Hamiltonian is

$$H = H_D + H_{DM} + H_{ND} + H_N \,. \tag{1}$$

Here $H_D$, $H_{DM}$, $H_{ND}$, and $H_N$ respectively represent the QD, the coupling between QD and MZM, the coupling between QD and normal lead, and the normal lead [42, 43, 46, 48, 49]:

$$H_D = (\epsilon_0 - V_Z)d_\uparrow^\dagger d_\uparrow + (\epsilon_0 + V_Z)d_\downarrow^\dagger d_\downarrow + U n_\uparrow n_\downarrow \,, \tag{2}$$

$$H_{DM} = it(d_\uparrow + d_\uparrow^\dagger)\gamma \,, \tag{3}$$

$$H_{ND} = \sum_{k\sigma} t_N c_{k\sigma}^\dagger d_\sigma + h.c. \,, \tag{4}$$

$$H_N = \sum_{k\sigma} \epsilon_k c_{k\sigma}^\dagger c_{k\sigma} \,, \tag{5}$$

where $d_\sigma$ and $c_{k\sigma}$ are annihilation operators of electrons in QD and normal lead, respectively, with spin $\sigma = \uparrow, \downarrow$. $\epsilon_0$ is the intra-dot energy level of the QD. $t_N$ is the hopping strength between the normal lead and the QD. The electron-electron interaction is included in $H_D$ as the term $U n_\uparrow n_\downarrow$, with $U$ the charging energy and $n_\sigma = d_\sigma^\dagger d_\sigma$ the particle number operator [1, 4, 6, 50–54]. In our calculations, we always set $U = 1$ as the energy unit. $\gamma$ is the operator of the MZM. The MZMs always emerge in pair, and their coupling strength is determined by the overlap of their wavefunctions [16, 22, 48]. Their nonlocality relates to separation between a pair of MZMs, which can be measured also through the normal lead-QD-MZM systems [22, 55]. In our study, we consider the pair of MZMs are well separated (e.g. as long as the Majorana nanowire is long). They are almost decoupled and only one MZM $\gamma$ couples to the QD [42, 43]. The generation of MZM usually demands the existence of a conventional superconductor, which is not included in our model. This is because we focus on the MZM at zero energy, where the DOS is not affected by the superconducting continuum outside the gap. Also, this is to provide a concise comparison to QD-superconductor systems.

In order to regulate the topological superconductor to the nontrivial phase, a magnetic term, such as an external magnetic field [18, 19] or magnetic exchange coupling of QD [8, 9], is usually demanded. Therefore, the QD inevitably feels a Zeeman energy $V_Z$, which here represents the effective magnetic field parallel to the spin-up direction. Due to the self-Hermitian property $\gamma^\dagger = \gamma$, the MZM couples to electrons and holes with the same strength $t$ [42], and only one spin channel is coupled [43]. Inducing the MZM usually demands a large Zeeman term $-V_Z\sigma_z$, and in this case the MZM almost just couples to spin $+z$ [22, 43, 44]. This corresponds to Eq. (3) where $\gamma$ is just coupled to the $d_\uparrow$ channel. Note that there are only two spin-dependent terms in the Hamiltonian: the Zeeman term and the MZM-QD coupling. Even if $V_Z$ is not large and the MZM couples to both $+z$ and $-z$ spin $ad_z + bd_{\bar{z}}$ ($a, b$ are normalized coefficients), one can rotate the spin basis as $d_\uparrow = ad_z + bd_{\bar{z}}$. In this new basis, the MZM still only couples to $d_\uparrow$, and the spin direction of the Zeeman term is a bit deviated from $\uparrow$ spin direction. If $V_Z = 0$ and the Zeeman term is absent, setting that MZM just couples to $d_\uparrow$ has no influence on any other term of the Hamiltonian. Therefore, it is reasonable to set that the MZM couples to electrons and holes of spin-up channel, as shown in Eq. (3).

In fact, when the normal lead is decoupled, the system can be exactly solved by diagonalization. Here we consider the normal lead coupled to the QD, because (i) a lead is usually needed to probe the existence of MZMs in experiments and (ii) the normal lead can facilitate the visualization of DOS by directly providing a broadening via the imaginary parts of retarded Green's functions. This broadening is important to reflect the MZM signal change along with

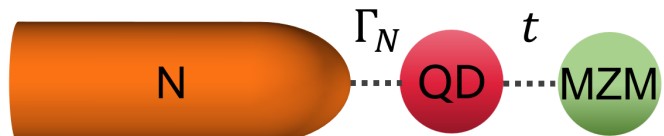

Figure 1: The schematic plot for the QD-MZM coupling system. In addition, the QD is weakly coupled to a normal lead, for the visualization of DOS and a better description of practical experiments. $\Gamma_N$ and $t$ respectively indicate the strength of QD-normal lead coupling and QD-MZM coupling.

the phase transition, as explained in Appendix A. The normal lead-QD coupling strength is described by $\Gamma_N = \pi \rho_N t_N^2$ with $\rho_N$ the DOS in the normal lead [46]. In normal lead-QD-MZM systems, there is also the Kondo effect, which has been studied by researchers [56, 57]. It corresponds to the case that the temperature $T$ is comparable to the Kondo temperature $T_K$. Below we set a weak normal lead-QD coupling with $\Gamma_N = 0.01U$. What is more, because the coupling between QD and normal lead is weak, the Kondo temperature $T_K$ is very low [58], and the condition $T >> T_K$ is easily met, thus our study is not relevant to the Kondo regime [56, 57, 59–61].

Below we first diagonalize the system without normal lead to obtain the GS. By doing this, the energy level and occupation numbers $\langle n_\sigma \rangle$ are exactly solved, and the phase transitions are revealed. Based on the GS, we introduce the normal lead as the imaginary part of Green's function, so that the DOS has a broadening and can be visualized. We represent the MZM by the normal Fermion operator $\gamma = \frac{1}{\sqrt{2}}(c + c^\dagger)$.

When the normal lead is absent, there exist four possible occupations of the QD, and two possible occupations of the MZM system. Therefore, the Hamiltonian can be written as a $8 \times 8$ matrix, in the basis $(|0, 0, 0\rangle, |1, 1, 0\rangle, |1, 0, 0\rangle, |0, 1, 0\rangle, |0, 0, 1\rangle, |1, 1, 1\rangle, |1, 0, 1\rangle, |0, 1, 1\rangle)$. Here

$$|i, j, k\rangle = |n_c = i, n_\uparrow = j, n_\downarrow = k\rangle = (c^\dagger)^i (d_\uparrow^\dagger)^j (d_\downarrow^\dagger)^k |0\rangle. \tag{6}$$

The Hamiltonian has four $2 \times 2$ blocks $H_1 \oplus H_2 \oplus H_3 \oplus H_4$, with

$$H_1 = \begin{pmatrix} 0 & \frac{it}{\sqrt{2}} \\ \frac{-it}{\sqrt{2}} & \epsilon_0 - V_Z \end{pmatrix}, \qquad H_2 = \begin{pmatrix} 0 & \frac{-it}{\sqrt{2}} \\ \frac{it}{\sqrt{2}} & \epsilon_0 - V_Z \end{pmatrix}, \tag{7}$$

$$H_3 = \begin{pmatrix} \epsilon_0 + V_Z & \frac{it}{\sqrt{2}} \\ \frac{-it}{\sqrt{2}} & 2\epsilon_0 + U \end{pmatrix}, \qquad H_4 = \begin{pmatrix} \epsilon_0 + V_Z & \frac{-it}{\sqrt{2}} \\ \frac{it}{\sqrt{2}} & 2\epsilon_0 + U \end{pmatrix}. \tag{8}$$

The four blocks correspond to eight eigenvalues

$$\epsilon_{1,\pm} = \epsilon_{2,\pm} = \frac{\epsilon_0 - V_Z \pm \sqrt{(\epsilon_0 - V_Z)^2 + 2t^2}}{2}, \tag{9}$$

$$\epsilon_{3,\pm} = \epsilon_{4,\pm} = \frac{3\epsilon_0 + U + V_Z \pm \sqrt{(\epsilon_0 + U - V_Z)^2 + 2t^2}}{2}. \tag{10}$$

Focusing on the occupation of the QD, we can find that $H_1, H_2$ both correspond to basis $(|n_\uparrow = 0, n_\downarrow = 0\rangle, |n_\uparrow = 1, n_\downarrow = 0\rangle)$, and $H_3, H_4$ both correspond to basis $(|n_\uparrow = 0, n_\downarrow = 1\rangle, |n_\uparrow = 1, n_\downarrow = 1\rangle)$. What is more, because $H_1 = H_2^*, H_3 = H_4^*$, their eigenvectors satisfy $\psi_{1,\pm} = \psi_{2,\pm}^*, \psi_{3,\pm} = \psi_{4,\pm}^*$. For the above reasons, $\psi_{1,\pm}$ and $\psi_{2,\pm}$ ($\psi_{3,\pm}$ and $\psi_{4,\pm}$), the degenerate eigenstates of $H_1$ and $H_2$ ($H_3$ and $H_4$), have the same occupations of the QD and indicate spin-up (spin-down) states. Thus, we can just analyze $H_1$ and $H_3$ only.

The GS energy can only equal to $\epsilon_{1,-}$ or $\epsilon_{3,-}$. The GS is judged by the sign of $\epsilon_{3,-} - \epsilon_{1,-}$. For $\epsilon_{1,-} < \epsilon_{3,-}$, the GS energy is $\epsilon_{1,-}$, and its occupation numbers can be obtained from $\psi_{1,-}$ for Hamiltonian $H_1$

$$\langle n_\uparrow \rangle = \frac{1}{2}\left(1 - \frac{\epsilon_0 - V_Z}{\sqrt{(\epsilon_0 - V_Z)^2 + 2t^2}}\right), \qquad \langle n_\downarrow \rangle = 0. \tag{11}$$

Because $\langle n_\downarrow \rangle = 0$, the state of the QD is spin-up and contributed by $|0\rangle$ and $|\uparrow\rangle$. For $\epsilon_{1,-} > \epsilon_{3,-}$, the GS energy is $\epsilon_{3,-}$, and its occupation numbers can be obtained from $\psi_{3,-}$ for Hamiltonian $H_3$

$$\langle n_\uparrow \rangle = \frac{1}{2}\left(1 - \frac{\epsilon_0 + U - V_Z}{\sqrt{(\epsilon_0 + U - V_Z)^2 + 2t^2}}\right), \qquad \langle n_\downarrow \rangle = 1. \tag{12}$$

Because $\langle n_\downarrow \rangle = 1$, the state is spin-down and contributed by $|\downarrow\rangle$ and $\frac{1}{\sqrt{2}}(|\uparrow\downarrow\rangle - |\downarrow\uparrow\rangle)$. When the parameters change, the sign of $\epsilon_{3,-} - \epsilon_{1,-}$ can also change and result in the GS transition between $\psi_{1,-}$ and $\psi_{3,-}$.

Next we solve the single-particle DOS from retarded Green's function. The single particle can be electron $e\sigma$ or hole $h\sigma$, with spin $\sigma = \uparrow, \downarrow$. The energy space Green's function is obtained from the time space via Fourier transformation

$$G^r_{D,e(h)\sigma e(h)\sigma}(\epsilon) = \int dt\, e^{i\epsilon t} G^r_{D,e(h)\sigma e(h)\sigma}(t). \tag{13}$$

The time-space Green's function of $e\sigma$ is

$$
\begin{aligned}
G^r_{D,e\sigma e\sigma}(t) &= -i\theta(t)\langle g|d_\sigma(t)d^\dagger_\sigma(0) + d^\dagger_\sigma(0)d_\sigma(t)|g\rangle \\
&= -i\theta(t)\sum_j [\langle g|d_\sigma(t)|j\rangle\langle j|d^\dagger_\sigma(0)|g\rangle + \langle g|d^\dagger_\sigma(0)|j\rangle\langle j|d_\sigma(t)|g\rangle] \\
&= -i\theta(t)\sum_j [e^{i(\epsilon_g - \epsilon_j)t}\langle g|d_\sigma(0)|j\rangle\langle j|d^\dagger_\sigma(0)|g\rangle + e^{i(\epsilon_j - \epsilon_g)t}\langle g|d^\dagger_\sigma(0)|j\rangle\langle j|d_\sigma(0)|g\rangle] \\
&= -i\theta(t)\sum_j [e^{i(\epsilon_g - \epsilon_j)t}|\widetilde{A}_\sigma(g,j)|^2 + e^{i(\epsilon_j - \epsilon_g)t}|\widetilde{A}_\sigma(j,g)|^2].
\end{aligned}
\tag{14}
$$

Here we use the eigenstate basis $j = 1, 2, 3, ..., 8$ corresponding to $\psi_{1,-}, \psi_{1,+}, \psi_{2,-}, ..., \psi_{4,+}$. $g$ indicates the order number $j$ of GS. When $\epsilon_{1,-} < \epsilon_{3,-}$ ($\epsilon_{1,-} > \epsilon_{3,-}$), $g = 1 (g = 5)$ indicates $\psi_{1,-}$ ($\psi_{3,-}$). Note that the term $\langle g|d_\sigma(t)|j\rangle$ is in the Heisenberg representation and can be transformed to Schrödinger representation $\langle g(t)|d_\sigma(0)|j(t)\rangle = e^{i(\epsilon_g - \epsilon_j)t}\langle g|d_\sigma(0)|j\rangle$. Similarly, we get $\langle j|d_\sigma(t)|g\rangle = e^{i(\epsilon_j - \epsilon_g)t}\langle j|d_\sigma(0)|g\rangle$. $\widetilde{A}_\sigma(x,y) = \langle x|d_\sigma(0)|y\rangle$ is the representation of $d_\sigma(0)$ in the $j$ basis. It is obtained by a unitary transformation on $A_\sigma$, which is the representation of $d_\sigma(0)$ in the basis of $H_1$ to $H_4$ (basis Eq. (6)): $A_\uparrow$ is a $8 \times 8$ matrix with four nonzero elements $A_\uparrow(1,4) = A_\uparrow(5,8) = 1$, $A_\uparrow(3,2) = A_\uparrow(7,6) = -1$. $A_\downarrow$ is also a $8 \times 8$ matrix with four nonzero elements $A_\downarrow(1,5) = A_\downarrow(2,6) = 1$, $A_\downarrow(3,7) = A_\downarrow(4,8) = -1$. The transformation is $\widetilde{A}_\sigma = V^\dagger A_\sigma V$, with $V = V_1 \oplus V_2 \oplus V_3 \oplus V_4$ obtained from the eigenvectors of $H_1$ to $H_4$:

$$V_1 = \begin{pmatrix} \frac{it}{\sqrt{t^2 + 2\epsilon^2_{1,-}}} & \frac{it}{\sqrt{t^2 + 2\epsilon^2_{1,+}}} \\ \frac{\sqrt{2}\epsilon_{1,-}}{\sqrt{t^2 + 2\epsilon^2_{1,-}}} & \frac{\sqrt{2}\epsilon_{1,+}}{\sqrt{t^2 + 2\epsilon^2_{1,+}}} \end{pmatrix}, \tag{15}$$

$$V_2 = \begin{pmatrix} \frac{-it}{\sqrt{t^2 + 2\epsilon^2_{1,-}}} & \frac{-it}{\sqrt{t^2 + 2\epsilon^2_{1,+}}} \\ \frac{\sqrt{2}\epsilon_{1,-}}{\sqrt{t^2 + 2\epsilon^2_{1,-}}} & \frac{\sqrt{2}\epsilon_{1,+}}{\sqrt{t^2 + 2\epsilon^2_{1,+}}} \end{pmatrix}, \tag{16}$$

$$V_3 = \begin{pmatrix} \frac{it}{\sqrt{t^2+2(\epsilon_{3,-}-\epsilon_0-V_Z)^2}} & \frac{it}{\sqrt{t^2+2(\epsilon_{3,+}-\epsilon_0-V_Z)^2}} \\ \frac{\sqrt{2}(\epsilon_{3,-}-\epsilon_0-V_Z)}{\sqrt{t^2+2(\epsilon_{3,-}-\epsilon_0-V_Z)^2}} & \frac{\sqrt{2}(\epsilon_{3,+}-\epsilon_0-V_Z)}{\sqrt{t^2+2(\epsilon_{3,+}-\epsilon_0-V_Z)^2}} \end{pmatrix}, \tag{17}$$

$$V_4 = \begin{pmatrix} \frac{-it}{\sqrt{t^2+2(\epsilon_{3,-}-\epsilon_0-V_Z)^2}} & \frac{-it}{\sqrt{t^2+2(\epsilon_{3,+}-\epsilon_0-V_Z)^2}} \\ \frac{\sqrt{2}(\epsilon_{3,-}-\epsilon_0-V_Z)}{\sqrt{t^2+2(\epsilon_{3,-}-\epsilon_0-V_Z)^2}} & \frac{\sqrt{2}(\epsilon_{3,+}-\epsilon_0-V_Z)}{\sqrt{t^2+2(\epsilon_{3,+}-\epsilon_0-V_Z)^2}} \end{pmatrix}. \tag{18}$$

From the process above, $\widetilde{A}_\sigma$ and $G^r_{D,e\sigma e\sigma}(t)$ are solved, and $G^r_{D,e\sigma e\sigma}(\epsilon)$ is obtained via Eq. (13)

$$G^r_{D,e\sigma e\sigma}(\epsilon) = \sum_j \left[ \frac{|\widetilde{A}_\sigma(g,j)|^2}{\epsilon - \epsilon_j + \epsilon_g + i\Gamma_N} + \frac{|\widetilde{A}_\sigma(j,g)|^2}{\epsilon - \epsilon_g + \epsilon_j + i\Gamma_N} \right]. \tag{19}$$

Here, the coupling of normal lead is included as the imaginary part $\Gamma_N = \pi\rho_N t_N^2 = 0.01U$ [46]. Similarly, $G^r_{D,h\sigma h\sigma}(\epsilon)$ can be solved by substituting $d_\sigma$ by $d_\sigma^\dagger$ in Eq. (14), and is equivalent to substituting $\widetilde{A}_\sigma$ by $\widetilde{A}_\sigma^\dagger$ in Eq. (19). The single-particle DOS is obtained from the retarded Green's function [4]

$$\rho_{e(h)\sigma}(\epsilon) = -\frac{1}{\pi} Im[G^r_{D,e(h)\sigma e(h)\sigma}(\epsilon)]. \tag{20}$$

# 3  Phase transition without Zeeman term

First we consider a simple case that the Zeeman term $V_Z = 0$. When the QD-MZM coupling strength $t = 0$, the result returns to that of an isolated QD [4–7, 11, 12]: The QD has a degenerate doublet GS in the range $-U < \epsilon_0 < 0$, while the GS is singlet outside this region. The physics we most concern with is how the doublet state of QD is influenced by the MZM, i.e. the case $-U < \epsilon_0 < 0$. When the MZM is not coupled to the QD ($t = 0$), the spin rotation symmetry leads to the doublet state: With the total occupation number being 1, the two degenerate states $|\uparrow\rangle$ and $|\downarrow\rangle$ are respectively occupied by just a spin-up electron and just a spin-down electron. The GS can be either $|\uparrow\rangle$ with $\langle n_\uparrow \rangle = 1, \langle n_\downarrow \rangle = 0$ or $|\downarrow\rangle$ with $\langle n_\uparrow \rangle = 0, \langle n_\downarrow \rangle = 1$.

The MZM only couples to spin-up channel with strength $t$, causing the broken spin rotation symmetry and broken degeneracy of doublet state. According to Eqs. (11,12), the two eigenstates $\psi_{1,-}, \psi_{3,-}$ consist of both spin-up and spin-down occupation. They are respectively dominated by spin-up and spin-down components, and can be respectively called spin-up state and spin-down state.

On this condition, the energies of spin-up state and spin-down state $\epsilon_{1,-}, \epsilon_{3,-}$ are different, and the GS is determined by the sign of

$$\epsilon_{3,-} - \epsilon_{1,-} = \frac{1}{2}\left[ 2\epsilon_0 + U + \sqrt{\epsilon_0^2 + 2t^2} - \sqrt{(\epsilon_0+U)^2 + 2t^2} \right]$$

$$= \frac{2\epsilon_0 + U}{2}\left[ 1 - \frac{U}{\sqrt{\epsilon_0^2 + 2t^2} + \sqrt{(\epsilon_0+U)^2 + 2t^2}} \right]. \tag{21}$$

Note that when $t \neq 0$, $\sqrt{\epsilon_0^2 + 2t^2} + \sqrt{(\epsilon_0+U)^2 + 2t^2} > U$, and $1 - \frac{U}{\sqrt{\epsilon_0^2 + 2t^2} + \sqrt{(\epsilon_0+U)^2 + 2t^2}} > 0$. Therefore, the sign of $\epsilon_{3,-} - \epsilon_{1,-}$ is determined by the sign of $2\epsilon_0 + U$. When $\epsilon_0 < -U/2$ ($\epsilon_0 > -U/2$), $\epsilon_{3,-} < \epsilon_{1,-}$ ($\epsilon_{3,-} > \epsilon_{1,-}$), the GS is the spin-down state $\psi_{3,-}$ (spin-up state $\psi_{1,-}$). As shown in Fig. 2(a), we calculate and compare the energy $\epsilon_{1,-}, \epsilon_{3,-}$, so that we judge which is the GS. Then the spin of GS $\langle n_\uparrow \rangle - \langle n_\downarrow \rangle$ is plotted in the $\epsilon_0, t$ parameter space. A remarkable signature is the phase transition at $\epsilon_0 = -U/2$, consistent with Eq. (21). Indeed, the GS is

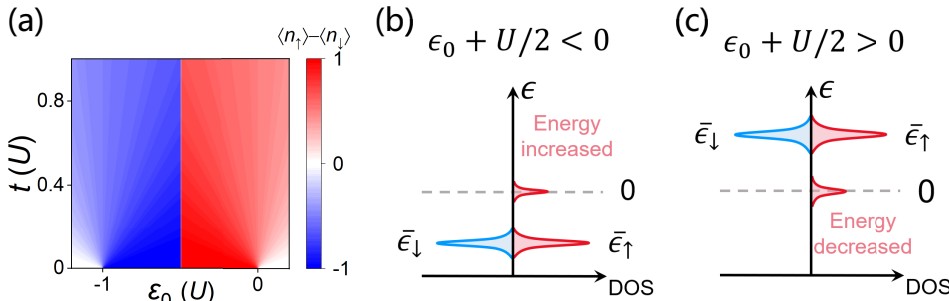

Figure 2: (a) The phase diagram versus intra-dot energy level $\epsilon_0$ and MZM coupling strength $t$ for $V_Z = 0$. Here we plot $\langle n_\uparrow \rangle - \langle n_\downarrow \rangle$ to show the spin polarization. (b, c) The mean-field picture for the phase transition. Without the coupling of MZM, spin $\uparrow$ and $\downarrow$ have the same average energy level $\bar{\epsilon}_\uparrow = \bar{\epsilon}_\downarrow = \epsilon_0 + U/2$. The leakage of MZM induces a zero-energy peak in spin-$\uparrow$ channel. Thus, the spin-$\uparrow$ energy is effectively increased (decreased) for $\epsilon_0 + U/2 < 0$ ($\epsilon_0 + U/2 > 0$), corresponding to a spin-down (spin-up) GS.

spin-down for $\epsilon_0 < -U/2$ and reversed to spin-up for $\epsilon_0 > -U/2$. The case is different from coupling to conventional superconductor, where the doublet GS can be changed to spin-singlet GS [4–7, 11, 12].

The phase transition can be understood by the single-particle effective energy levels in a mean-field picture. Due to the intra-dot Coulomb repulsion $U n_\uparrow n_\downarrow$, the energy level of certain spin is lifted from $\epsilon_0$ by the filled electron with opposite spin: The spin-up and spin-down occupations are determined by their spin-dependent effective energy levels $\epsilon_\uparrow = \epsilon_0 + \langle n_\downarrow \rangle U$ and $\epsilon_\downarrow = \epsilon_0 + \langle n_\uparrow \rangle U$. Without coupling of MZM ($t = 0$) and for doublet state ($-U < \epsilon_0 < 0$), the spin-up state $|\uparrow\rangle$ corresponds to $\langle n_\uparrow \rangle = 1$ and $\langle n_\downarrow \rangle = 0$, so $\epsilon_\uparrow = \epsilon_0$ and $\epsilon_\downarrow = \epsilon_0 + U$ are respectively below and above the Fermi energy $E_F = 0$. Self-consistently, these spin-dependent effective levels indicate occupation numbers $\langle n_\uparrow \rangle = 1$ and $\langle n_\downarrow \rangle = 0$ and that only spin-up channel is occupied [1, 4, 6]. Similarly, the spin-down state $|\downarrow\rangle$ corresponds to $\epsilon_\uparrow = \epsilon_0 + U$ and $\epsilon_\downarrow = \epsilon_0$. The discussion and symbol $\epsilon_\sigma$ above is based on that GS spin has been determined. Before the GS is determined, we consider both cases of spin polarization and take the average. The average energy levels are $\bar{\epsilon}_\uparrow = \bar{\epsilon}_\downarrow = \epsilon_0 + U/2$, as schematically shown in the two major DOS peaks in Figs. 2(b, c). Therefore, the GS is degenerate doublet state $|\uparrow\rangle$ and $|\downarrow\rangle$.

When MZM is coupled to QD with $t \neq 0$, the MZM leaks into the spin-up channel of the QD [62], bringing an additional peak at zero energy [zero-energy peaks in Figs. 2(b, c)]. The spin-up channel is initially located at $\bar{\epsilon}_\uparrow$, the MZM induced zero-energy peak effectively moves its energy level close to 0. When $\bar{\epsilon}_\uparrow = \bar{\epsilon}_\downarrow = \epsilon_0 + U/2 < 0$, the effective energy level of spin up is lifted to higher than $\bar{\epsilon}_\downarrow$, as shown in Fig. 2(b). The higher energy of the spin-up channel indicates that the GS is the spin-down state $\psi_{3,-}$. On the other hand, for $\epsilon_0 + U/2 > 0$, the spin-up energy is effectively reduced by MZM coupling, as shown in Fig. 2(c). Thus, the spin-up channel has the lower energy than spin-down channel, and the GS is spin-up state $\psi_{1,-}$. This picture explains the phase transition and spin change in Eq. (21) and Fig. 2(a).

In the presence of QD-MZM coupling $t$, the broken spin rotation symmetry not only destroys the degeneracy of doublet state for $-U < \epsilon_0 < 0$, but also transforms the initial spin-singlet state for $\epsilon_0 < -U$ or $\epsilon_0 > 0$ to become spin polarized. In other words, the GS is spin-polarized in the whole phase diagram [Fig. 2(a)], which is distinct from the doublet-singlet phase diagram in spin-singlet superconductor-QD system [4–7, 11, 12]. In addition, if the MZM is decoupled, the QD should be occupied by zero or two electrons when $\epsilon_0 > 0$ or $\epsilon_0 < -U$, and the state should respectively be the spin-singlet $|0\rangle$ or $\frac{1}{\sqrt{2}}(|\uparrow\downarrow\rangle - |\downarrow\uparrow\rangle)$. Thus, the corresponding spin polarization is nearly zero for very small MZM coupling $t$.

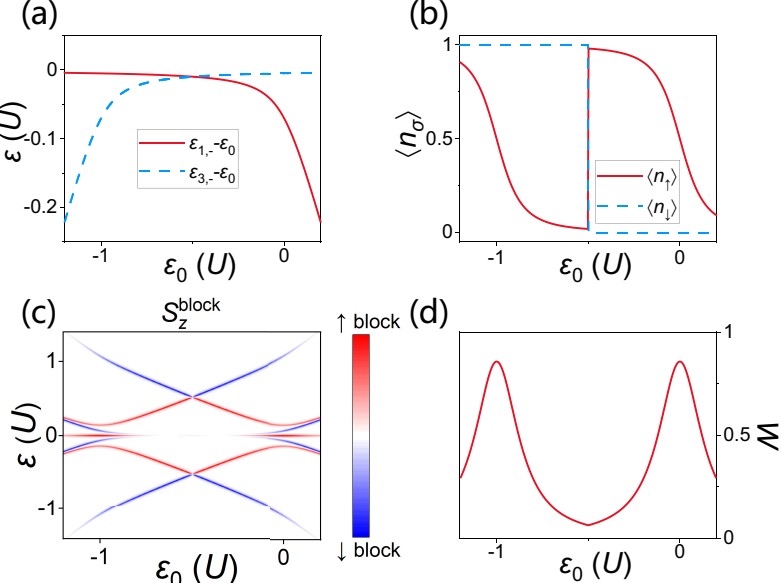

Figure 3: Phase transition of GS versus intra-dot energy level $\epsilon_0$ for $V_Z = 0$. (a) Energy comparison of spin-up and spin-down states $\epsilon_{1,-}$ and $\epsilon_{3,-}$. $\epsilon_0$ is subtracted for clarity. (b) The occupation numbers $\langle n_\uparrow \rangle$, $\langle n_\downarrow \rangle$ of GS. (c) The spin-resolved single-particle DOS. (d) The weight of zero-energy spin-up DOS. In these figures (a-d), the QD-MZM coupling strength $t = 0.1U$.

We also investigate the features of GS phase transition versus the intra-dot energy level $\epsilon_0$. In experiments this $\epsilon_0$ can be regulated by applying a gate voltage [4–6, 21]. The QD-MZM coupling strength is fixed to be $t = 0.1U$. The energy comparison of states $\psi_{1,-}, \psi_{3,-}$ is plotted in Fig. 3(a). Because $\epsilon_{1,-}, \epsilon_{3,-}$ are both mainly proportional to $\epsilon_0$, the energies are simultaneously subtracted by $\epsilon_0$ in Fig. 3(a) for a clear comparison. Just as the Eq. (21) and Fig. 2(a), $\epsilon_{1,-} > \epsilon_{3,-}$ ($\epsilon_{1,-} < \epsilon_{3,-}$) for $\epsilon_0 + U/2 < 0$ ($\epsilon_0 + U/2 > 0$), indicating the GS is the spin-down (spin-up) state. Fig. 3(b) shows the occupation numbers $\langle n_\uparrow \rangle$, $\langle n_\downarrow \rangle$ versus $\epsilon_0$. As $\epsilon_0$ increases and crosses $-U/2$ and phase transition happens, the spin polarization of GS undergoes a sharp transition from $\langle n_\downarrow \rangle = 1$ to $\langle n_\downarrow \rangle = 0$. In the mean-field picture, $\epsilon_\uparrow = \epsilon_0 + \langle n_\downarrow \rangle U$ also changes from $\epsilon_\uparrow = \epsilon_0 + U = 0.5U$ to $\epsilon_\uparrow = \epsilon_0 = -0.5U$. For $\epsilon_\uparrow = 0.5U > 0$, the spin-up channel is almost not occupied, with $\langle n_\uparrow \rangle \approx 0$. But for $\epsilon_\uparrow = -0.5U < 0$, the spin-up channel is almost occupied, with $\langle n_\uparrow \rangle \approx 1$. On the other hand, the MZM-induced zero-energy peak tends to move $\langle n_\uparrow \rangle$ to 0.5, thus around $\epsilon_0 = -U/2$, $\langle n_\uparrow \rangle$ is a bit deviated from 0 or 1. The lower $|\epsilon_\uparrow|$ is, the more evident the MZM-induced zero-energy leakage. Because $\epsilon_\uparrow = \pm U/2$ is far from zero, the leakage effect is weak and $\langle n_\uparrow \rangle$ is almost 0 or 1 around $\epsilon_0 = -U/2$. For the two separated regions $\epsilon_0 < -U/2, \epsilon_0 > -U/2$, as $\epsilon_0$ increases, $\epsilon_\uparrow$ increases and $\langle n_\uparrow \rangle$ decreases, while the decrease is not sharp due to the MZM coupling, as shown in Fig. 3(b).

Next we study its single-particle DOS. As shown in Fig. 3(c), we plot the spin-resolved DOS, which is defined as [63]

$$S_z^{\text{block}} = \rho_{e\uparrow} - \rho_{e\downarrow} + \rho_{h\uparrow} - \rho_{h\downarrow}. \tag{22}$$

Here we set $e\uparrow, h\uparrow$ components as positive (red color), and set $e\downarrow, h\downarrow$ components as negative (blue color). This quantity also reflects the total single-particle DOS. For $t = 0$ without MZM, the DOS of doublet state is a Coulomb diamond centered at $\epsilon_0 = -U/2$, like shown in experiments [4, 13]: Two electron levels $\epsilon_{e1} = \epsilon_0, \epsilon_{e2} = \epsilon_0 + U$ and two hole levels $\epsilon_{h1} = -\epsilon_0, \epsilon_{h2} = -\epsilon_0 - U$, intersecting at points $(\epsilon_0, \epsilon) = (-U, 0), (0, 0), (-U/2, U/2)$, and $(-U/2, -U/2)$. As MZM is coupled to QD, the Coulomb diamond shape almost keeps, but has

two differences: First, at $\epsilon_0 = -U/2$ the electron spin is reversed due to phase transition, see the two electron-like levels $\epsilon_{e1} \approx \epsilon_0, \epsilon_{e2} \approx \epsilon_0 + U$ in Fig. 3(c). Because $\epsilon_0 = -U/2$ is the particle-hole symmetric point, the phase transition just changes the signs of levels, and there is not a sharp change in the total DOS spectrum. Second, the spectrum opens two gaps at $\epsilon_0 = -U, 0$. Inside the gaps, the zero-energy positive peak is apparent. Because the MZM couples to spin-up channel, this peak indicates the high equal-spin Andreev reflection strength, which is a symbolic signature of the MZM [43, 63].

To quantitatively show the MZM signal, we calculate the weight of the zero-energy peak presented in Fig. 3(d), which is defined as

$$W = \int_{-0.04U}^{0.04U} d\epsilon (\rho_{e\uparrow} + \rho_{h\uparrow}). \tag{23}$$

Because the MZM is only coupled to the spin-up channel, we only consider the DOS from $e\uparrow$-$h\uparrow$ block and exclude irrelevant contributions. The weight is high at $\epsilon_0 = -U, 0$, but low around $\epsilon_0 = -U/2$. The distinct MZM signal can also be understood from the mean-field picture. In fact, the MZM can always induce a zero energy peak as shown in Fig. 3(c), but the leakage strength is strongly dependent on the ratio $t/|\epsilon_\uparrow|$. Note that the leakage of MZM is strong for a low $|\epsilon_\uparrow|$ value. For $\epsilon_0 < -U/2$, $\epsilon_\uparrow = \epsilon_0 + U$ is zero at $\epsilon_0 = -U$. For $\epsilon_0 > -U/2$, $\epsilon_\uparrow = \epsilon_0$ is zero at $\epsilon_0 = 0$. Therefore, the weight is maximized at $\epsilon_0 = -U, 0$. If the spin-up effective level $\epsilon_\uparrow$ is far away from zero (e.g. $\epsilon_0 = -U/2$), the MZM will be prohibited from leaking into the QD. It indicates that when experimentally probing MZM, even if the MZM actually exists, its signal may be subtle because it is weakened by a high QD energy level $|\epsilon_\uparrow|$.

## 4  Phase transition with Zeeman term

Above we study the phase transition without considering the Zeeman term. In fact, this Zeeman term should be included, because the nontrivial phase of topological superconductors and MZM are usually induced by a magnetic field [18, 19], or by an exchange coupling from a magnetic QD [8, 9]. The magnetic direction is approximately parallel to the MZM coupling channel spin up [43, 44]. Below we study the case with a Zeeman term, which is always set as $V_Z = 0.06U$. By involving the practical Zeeman term, the phase transition features become remarkable and can be used to understand MZM-related experiments.

As the Zeeman term is involved, when $t = 0$, the degenerate doublet GS is destroyed to a spin-polarized GS by the Zeeman term, where the energies of the spin-up and spin-down states are split by $2V_Z$. The phase diagram versus $\epsilon_0$ and $t$ is shown as Fig. 4(a): Basically, the GS is spin-up for a high $\epsilon_0$, and is spin-down for a low $\epsilon_0$. However, the phase transition with $V_Z \neq 0$ does not happen at the particle-hole symmetry point $\epsilon_0 = -U/2$.

To understand this feature, we also use the mean-field picture Figs. 4(b, c). The spin-dependent effective energy levels are $\epsilon_\uparrow = \epsilon_0 + \langle n_\downarrow \rangle U - V_Z$ and $\epsilon_\downarrow = \epsilon_0 + \langle n_\uparrow \rangle U + V_Z$. When MZM is absent $t = 0$, by substituting $(\langle n_\uparrow \rangle, \langle n_\downarrow \rangle) = (1, 0), (0, 1)$ and taking the average, one finds $\bar{\epsilon}_\uparrow = \epsilon_0 + U/2 - V_Z$, $\bar{\epsilon}_\downarrow = \epsilon_0 + U/2 + V_Z$ that determine GS spin. The relation $\bar{\epsilon}_\uparrow < \bar{\epsilon}_\downarrow$ destroys the degeneracy of doublet GS to spin-up GS. The MZM can change the spin-up GS to spin-down through the leakage effect: As shown in Figs. 4(b, c), the MZM effectively lifts (reduces) the energy of spin-up state towards 0 for $\bar{\epsilon}_\uparrow < 0$ ($\bar{\epsilon}_\uparrow > 0$). Thus, if the effective energy of spin up is lifted to higher than $\bar{\epsilon}_\downarrow$, the GS will be changed to the spin-down state. This demands two conditions: First, $\bar{\epsilon}_\downarrow < 0$ (which sufficiently satisfies $\bar{\epsilon}_\uparrow < 0$) because the effective energy of spin up is at most raised to 0. Second, the QD-MZM coupling $t$ should be high enough, so that the spin-up energy can be lifted to overcome the energy difference $\bar{\epsilon}_\downarrow - \bar{\epsilon}_\uparrow = 2V_Z$. Note that for a low $\epsilon_0$, the ratio $|\bar{\epsilon}_\downarrow - \bar{\epsilon}_\uparrow|/|\bar{\epsilon}_\downarrow|$ is low, and the phase transition can

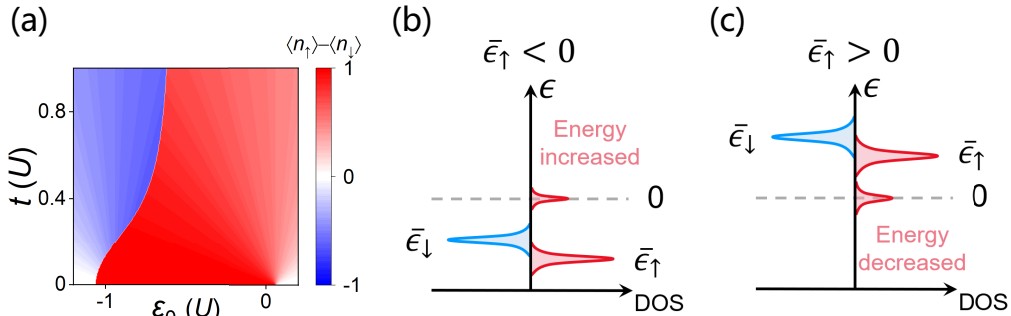

Figure 4: (a) The phase diagram versus intra-dot energy level $\epsilon_0$ and MZM coupling strength $t$ for $V_Z = 0.06U$. Here we plot $\langle n_\uparrow \rangle - \langle n_\downarrow \rangle$ to show the spin polarization. (b, c) The mean-field picture for the phase transition. Without the coupling of MZM, spin $\uparrow$ and $\downarrow$ have different average energy levels $\bar{\epsilon}_\uparrow = \epsilon_0 + U/2 - V_Z$, $\bar{\epsilon}_\downarrow = \epsilon_0 + U/2 + V_Z$. $\bar{\epsilon}_\uparrow < \bar{\epsilon}_\downarrow$ causes a spin-up GS. The MZM effectively lifts (decreases) the spin-$\uparrow$ energy for $\bar{\epsilon}_\uparrow < 0$ ($\bar{\epsilon}_\uparrow > 0$). When the effective spin-$\uparrow$ energy is lifted over spin-$\downarrow$ energy, the GS changes from spin-up to spin-down.

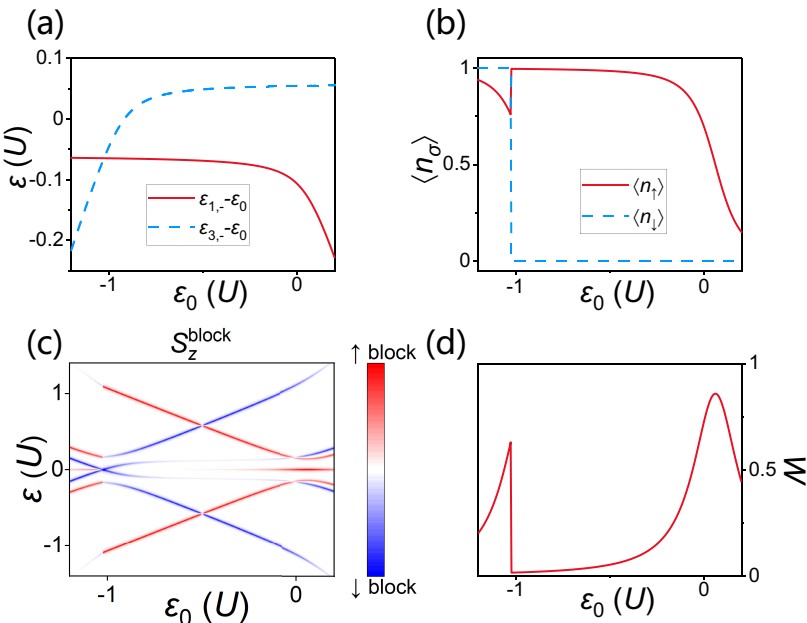

Figure 5: Phase transition of GS versus intra-dot energy level $\epsilon_0$ for $V_Z = 0.06U$. (a) Energy comparison of spin-up and spin-down states $\epsilon_{1,-}$ and $\epsilon_{3,-}$. $\epsilon_0$ is subtracted for clarity. (b) The occupation numbers $\langle n_\uparrow \rangle$, $\langle n_\downarrow \rangle$ of GS. (c) The spin-resolved single-particle DOS. (d) The weight of zero-energy spin-up DOS. In these figures (a-d), the QD-MZM coupling strength $t = 0.1U$.

happen for a relatively low QD-MZM coupling $t$, as shown in Fig. 4(a). The GS transition line is vertical without the Zeeman energy [Fig. 2(a)], which means that the GS can only change by regulating the intra-dot energy level $\epsilon_0$. But the Zeeman term changes the GS transition line to be oblique [Fig. 4(a)], and it becomes possible to also change the GS via just increasing QD-MZM coupling strength $t$, which is studied later.

The representative phase transition versus intra-dot energy level $\epsilon_0$ is summarized in Fig. 5, fixing $t = 0.1U$. Compared to the $V_Z = 0$ case Fig. 3(a), the energy of spin-up state $\epsilon_{1,-}$ and spin-down state $\epsilon_{3,-}$ is respectively reduced and lifted by about $V_Z$. This leads to the change of critical intra-dot energy level from $\epsilon_0 = -U/2$ to $\epsilon_0 = \epsilon_c < -U/2$ [Fig. 5(a)]. In Fig. 5(b), the occupation number $\langle n_\downarrow \rangle$ is suddenly changed from 1 to 0 at $\epsilon_0 = \epsilon_c$. But the change of $\langle n_\uparrow \rangle$ is not remarkable, because the critical energy level $\epsilon_c$ is about $-U$ and GS tends to be a double occupation singlet state $\frac{1}{\sqrt{2}}(|\uparrow\downarrow\rangle - |\downarrow\uparrow\rangle)$ and spin-up level always tends to be occupied.

With the Zeeman term, the single-particle DOS versus $\epsilon_0$ still behaves the Coulomb diamond feature, as shown in Fig. 5(c). Unlike the phase transition and spin reversion in Fig. 3(c), here the spin keeps in the range $\epsilon_0 > \epsilon_c \approx -U$ [see the spectral lines with positive slopes] indicating the large parameter range of the spin-up GS. When the phase transition happens ($\epsilon_0 = \epsilon_c$), the spin-down states intersect at zero energy. Meanwhile, the spin-resolved DOS peaks with nonzero energy have the energy unchanged but spin sign reversed. Notably, the zero-energy peak of MZM is subtle on the right of $\epsilon_c$, but is obvious on the left. This is because $\epsilon_\uparrow = \epsilon_c + U \approx 0$ on the left suddenly changes to $\epsilon_\uparrow = \epsilon_c \approx -U$ on the right. The sharp increase of $|\epsilon_\uparrow|$ causes the sharp decrease of MZM leakage, which is quantitatively shown in the weight $W$ [Fig. 5(d)]. This can be analogized to the weight transitions of Andreev bound states in QD-conventional superconductor system in Ref. [2]. In Fig. 5(d), with the increase of $\epsilon_0$ from $\epsilon_c$, the weight gradually becomes apparent due to the decreased $|\epsilon_\uparrow|$, and it has a large value for a high energy level $\epsilon_0$, like the $V_Z = 0$ case Fig. 3(d). Correspondingly, in Fig. 5(c) as $\epsilon_0$ is further increased to about 0, the MZM signal becomes apparent again. These results are similar to the experimental result by Mourik et al. in a Majorana nanowire (their Fig. 3A) [23]: A QD region is formed by a section of nanowire with the energy level controlled by the gate voltage, which corresponds to $-\epsilon_0$ in our work. When regulating the gate voltage, the nonzero-energy states cross at zero energy. Around the crossing, the zero energy signature seems missing on one side, but becomes apparent on the other side. Also, on the signature-missing side, as gate voltage is turned away from the crossing point, the zero energy peak gradually appears [23]. The zero-energy crossing, the sharp change of MZM signal, and the reemergence of zero-bias peak are very consistent with our results Figs. 5(c, d). Our theoretical analysis can provide such kind of experiments with a potential understanding from the perspective of QD phase transitions, with MZM already existed. They also indicate that even if the zero-bias peak is absent, we can not definitely judge that the MZM is absent.

As shown by the phase diagram Fig. 4(a), the phase transition can also occur by just increasing QD-MZM coupling strength $t$. For a fixed intra-dot energy level $\epsilon_0 = -0.9U$, increasing $t$ from zero to the critical value $t_c$ indeed leads to the phase transition. As shown in Fig. 6(a), the energies of spin-up and spin-down states are split by about $2V_Z$ at $t \to 0$, indicating a spin-up GS. Along with the increase of $t$, the energies of two states both decrease but the spin-down energy $\epsilon_{3,-}$ decreases faster. When $t$ reaches $t_c$, $\epsilon_{3,-}$ becomes lower than $\epsilon_{1,-}$ and the GS becomes the spin-down state $\psi_{3,-}$. In comparison, for $V_Z = 0$, when $t = 0$, $\epsilon_{3,-} = \epsilon_{1,-} = \epsilon_0$ are degenerate in the doublet region. For the same $\epsilon_0 = -0.9U$, due to the faster decrease of $\epsilon_{3,-}$ versus $t$, the GS becomes spin-down state as long as $t \neq 0$, consistent with the $V_Z = 0$ phase diagram Fig. 2(a).

The occupation numbers versus $t$ in Fig. 6(b) also show the phase transition. Along with the increase of $t$ and phase transition happens at $t = t_c$, $\langle n_\downarrow \rangle$ changes from 0 to 1, $\epsilon_\uparrow = \epsilon_0 + \langle n_\downarrow \rangle U$ changes from $-0.9U$ to $0.1U$. Because of the coupling of the MZM, the occupation number $\langle n_\uparrow \rangle$ always tends to be 0.5 as $t$ increases, for both $t < t_c$ and $t > t_c$. For $t < t_c$ and $\epsilon_\uparrow = -0.9U$, the spin-up channel is almost occupied with $\langle n_\uparrow \rangle \approx 1$. After phase transition $t > t_c$, $\epsilon_\uparrow = 0.1U$ is much smaller than the QD-MZM coupling $t$, so the MZM leakage turns $\langle n_\uparrow \rangle$ to be about 0.5. Therefore, the evolved state of the QD for a large $t$ is almost equally contributed by a spin-down state $\frac{1}{\sqrt{2}}|\downarrow\rangle$ and a double occupation singlet state $\frac{1}{2}(|\uparrow\downarrow\rangle - |\downarrow\uparrow\rangle)$.

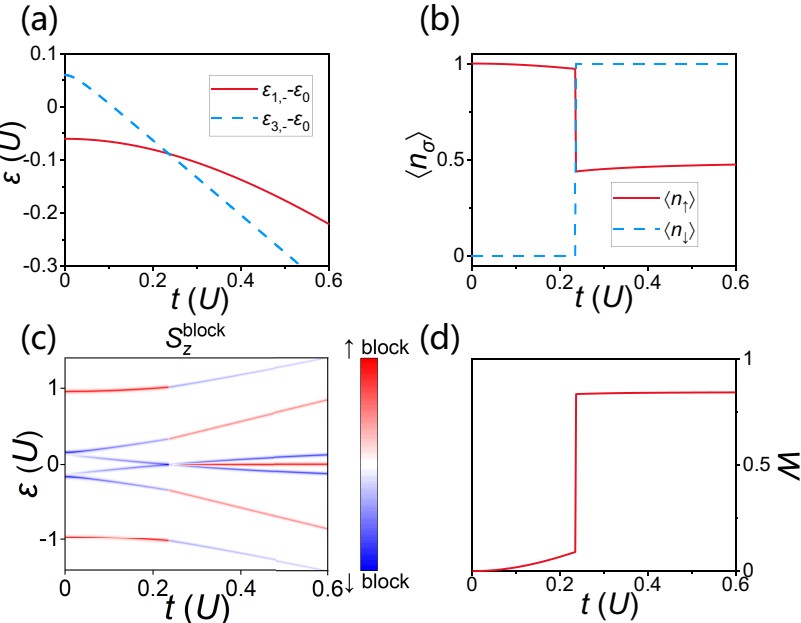

Figure 6: Phase transition of GS versus QD-MZM coupling strength $t$ for $V_Z = 0.06U$. (a) Energy comparison of spin-up and spin-down states $\epsilon_{1,-}$ and $\epsilon_{3,-}$. $\epsilon_0$ is subtracted for clarity. (b) The occupation numbers $\langle n_\uparrow \rangle$, $\langle n_\downarrow \rangle$ of GS. (c) The spin-resolved single-particle DOS. (d) The weight of zero-energy spin-up DOS. In these figures (a-d), the intra-dot energy level $\epsilon_0 = -0.9U$.

The spin-resolved single-particle DOS of the GS is also shown in Fig. 6(c). Like the phase transition versus $\epsilon_0$, the spin-down levels cross at zero energy at transition point $t = t_c$, and the nonzero-energy peaks have energy unchanged but spin sign reversed at $t = t_c$. The zero energy peak, which reflects the leakage of MZM, is subtle when $t < t_c$ but apparent when $t > t_c$, because $|\epsilon_\uparrow|$ is decreased from $0.9U$ to $0.1U$. The weight in Fig. 6(d) gives the quantitative description of the emergence of strong zero energy peak.

The MZM *becomes apparent only when the coupling strength $t$ reaches a critical value $t_c$ that leads to the phase transition.* Our theoretical result could provide an understanding of MZM-related transport measurements. It is consistent with the recent experimental work by Fan et al. in the platform of iron-based superconductor [9], which is believed as one of condensed matter systems to realize MZMs [9, 30, 31]. Some adatoms are deposited on the surface of the superconductor and create nearby MZMs via their exchange coupling. The adatom can be viewed as a QD, and its coupling strength to the MZM is controlled by the distance between the adatom and the superconductor surface. As the adatom is pushed toward the superconductor, the coupling strength increases and the nonzero energy states cross, and the MZM zero-energy peak appears after this crossing [9].

In phase transitions versus both intra-dot energy level $\epsilon_0$ and QD-MZM coupling strength $t$, the single-particle DOS Figs. 5(c), 6(c) exhibit energy level crossing at the transition point $\epsilon_c, t_c$. Also, after the phase transition, the MZM zero-energy peak becomes apparent, which may be mistakenly regarded as the emergence of MZM itself: Similarly, when researchers regulate topological transition and induce the appearance of MZM, *the energy gap usually closes, and reopens with a new zero-energy peak* indicating the MZM emergence [16, 33]. Here, we show that even when MZM already exists, the phase transition of QD leads to *the same feature* as that of topological transition. Therefore, the MZM does not necessarily induce a zero-bias peak. Even if the zero-bias peak does not exist, one can not definitely judge that the MZM is nonexistent.

## 5 Conclusion

In summary, the phase transitions in QD-MZM coupling systems are investigated. The phase diagrams without and with Zeeman terms are both given, showing the transition lines. The phase transitions can happen via regulating the intra-dot energy level or QD-MZM coupling strength. Along with these phase transitions, the occupation numbers and single-particle DOS are studied. The transition features can be understood by the mean-field picture. Our study not only provides an analogy to QD-superconductor phase transitions, but also offers an understanding on MZM-probing experiments.

## Acknowledgments

We are grateful to Yu-Chen Zhuang and Yi-Xin Dai for fruitful discussions.

**Funding information** This work was financially supported by the National Key R and D Program of China (Grant No. 2024YFA1409002), the National Natural Science Foundation of China (Grants No. 12374034 and No. 11921005), and the Innovation Program for Quantum Science and Technology (Grant No. 2021ZD0302403). The computational resources are supported by High-performance Computing Platform of Peking University.

## A The role of normal lead

In Eq. (19), the normal lead just gives a finite broadening to the states, which may seem to be just a complication in computation. However, this broadening is essential for demonstrating the change of MZM weight and the inspiration to experimental detections.

In Fig. 7(a), we plot the energy of eigenstates versus $\epsilon_0$. This corresponds to Fig. 5(c) and can be obtained no matter the normal lead coupling is present or not. For clarity, we also show Fig. 5(c) again in Fig. 7(b), which can be obtained only when the normal lead is coupled to the QD. Indeed, the eigenenergies Fig. 7(a) are consistent with the spin-resolved DOS Fig. 7(b). However, compared to Fig. 7(b), Fig. 7(a) lacks the weight information: In Fig. 7(a), one finds that a zero-energy state always exists. Only in Fig. 7(b) when the QD couples to normal lead, one can identify that the MZM weight changes violently versus $\epsilon_0$, and notice that the phase transition plays an important role on the visibility of MZM signal.

Therefore, the coupling of normal lead provides the weight information, which can not be obtained by just solving the eigenenergy. On the other hand, the normal lead is usually demanded in MZM detections, thus introducing the lead is natural and consistent with experimental conditions.

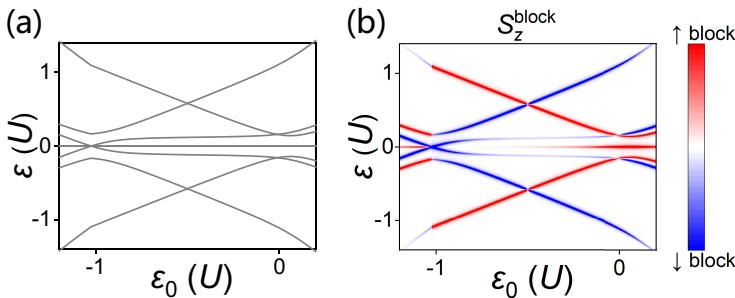

Figure 7: (a) The energy of states versus $\epsilon_0$. (b) In the presence of a normal lead, the spin-resolved single-particle DOS versus $\epsilon_0$ (the same data as Fig. 5(c), but the colorbar is adjusted for clarity).

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
