# Peer review of "Phase transitions in quantum dot-Majorana zero mode coupling systems"

_SciPost Physics Core, doi:SciPost Phys. Core 8, 031 (2025)_

## Round 1 · Referee Report · Anonymous (Referee 1) · 2024-8-21

Strengths

  1. Provides analytical solution

Weaknesses

  1. No discussion of relationship of results as compared to the large set of previous papers studying coupling of a quantum dot to a Majorana zero mode
  2. Model seems overly simplified
  3. No discussion of how to observe the effect in experiment
  4. Role of lead seems not to add anything relevant in terms of physics

Report

The paper deals with the problem of theoretically computing the ground state of a quantum dot coupled to a Majorana zero mode.

I do appreciate that the authors provide analytical solutions to this problem. There are however several major problems with this manuscript:

  1. There have been several publications earlier about coupling a Majorana bound state to a quantum dot, e.g. Refs. [22, 57] cited in the paper, or e.g. https://doi.org/10.1103/PhysRevB.96.201109 . The paper does not discuss at all how the results relate to any of the previous work.
  2. The model studied seems too simple for me: (i) in principle a Majorana bound state can couple to both spins in the quantum dot (see Ref. [22]), and (ii) the authors neglect the coupling to the superconducting continuum (at finite energy!). These simplifications are not discussed.
  3. I am also missing a discussion of experimental relevance of the findings. The authors talk about a phase transition, but in the end it's just about whether the ground state spin is up or down. What is the relevance of this?
  4. The authors introduce a normal lead. However, in the approximation they use, the lead just gives a finite broadening to the states. To me, it seems just a complication to compute the Green's function - the eigenstates give exactly the same physical information.

Unless these problems are addressed, I cannot recommend publishing the paper in any of SciPost's journals.

Recommendation

Ask for major revision

  • validity: low
  • significance: low
  • originality: low
  • clarity: low
  • formatting: mediocre
  • grammar: mediocre

Author:  Yue Mao  on 2024-09-05  [id 4739]

(in reply to Report 1 on 2024-08-21)

We thank the referee for carefully reading our manuscript and the valuable report. We provide our response to the questions raised by the referee in the attached file.

Attachment:

reply_mzm_pt_1_0905.pdf

---

## Round 2 · Referee Report · Anonymous (Referee 1) · 2025-1-21

Strengths
- provides analytical expressions
- clear explanation of what is done and what effects are neglected
Weaknesses
- potentially important effects are neglected, but the paper is now transparent about this
Report
The authors have answered my questions, and extended their discussions considerably. I still believe that the paper neglects what are relevant effects in real experiments. However, they authors are now also very transparent about this aspect.
Hence, I believe the acceptance criteria for Scipost Physics Core are fulfilled, given that the analytical results of the authors can be interesting to some practitioners in the field.
However, the paper has several language errors, and I would recommend to have it properly proofread.
Hence, I believe the acceptance criteria for Scipost Physics Core are fulfilled, given that the analytical results of the authors can be interesting to some practitioners in the field.
However, the paper has several language errors, and I would recommend to have it properly proofread.
Recommendation
Ask for minor revision

Author: Yue Mao on 2025-02-03 [id 5180]
(in reply to Report 1 on 2025-01-21)We thank the referee for the feedback and suggestions. We have carefully checked the language of our manuscript, and made modifications accordingly.

---

## Round 2 · Author Response

We thank the editor for taking care of our paper and considering it for the review process. We thank the referee for reading our manuscript and providing critical comments. We have provided the reply to comments raised by the referee as individual author reply.

---

## Round 2 · List of Changes

- At the end of abstract, we add a sentence to emphasize the importance of our study.
- In lines 71-74, we add discussions on previous studies about lead-QD-MZM systems.
- In lines 75-78, we additionally explain why the superconductor is not included.
- In lines 84-93, we additionally explain MZM-QD coupling.
- In lines 100-102, 105, we add discussions on previous studies about Kondo effect.
- In lines 299-311, 353-354, we add some contents to emphasize our study's relevance to experiments.
- In lines 370-385, we add an Appendix to clarify the role of normal lead. This is also additionally mentioned in lines 59, 98-99.
- We also add a figure [Fig. 7] to help clarifying the role of normal lead.
- We add a reference [Ref. 55].

---

## Round 3 · Author Response

We thank the editor for taking care of our paper and processing it. We thank the referee for the feedback, saying the ``acceptance criteria for Scipost Physics Core are fulfilled''. We also thank the referee's suggestion on revising our language errors. We have carefully checked and modified the language of our manuscript.

---

## Round 3 · List of Changes

• We revise the language problems, including spelling, grammar, and expressions. The changes are marked as red color in the manuscript.

---

## Editorial Decision

published